# Fatal Epileptic Seizures in Mice Having Compromised Glutathione and Ascorbic Acid Biosynthesis

**DOI:** 10.3390/antiox12020448

**Published:** 2023-02-10

**Authors:** Ying Chen, Katherine D. Holland, Howard G. Shertzer, Daniel W. Nebert, Timothy P. Dalton

**Affiliations:** 1Department of Environmental and Public Health Sciences, Center for Environmental Genetics, University of Cincinnati College of Medicine, Cincinnati, OH 45267, USA; 2Department of Environmental Health Sciences, Yale School of Public Health, Yale University, New Haven, CT 06520, USA; 3Division of Neurology, Cincinnati Children’s Hospital Medical Center, Cincinnati, OH 45229, USA; 4Department of Pediatrics, University of Cincinnati College of Medicine, Cincinnati, OH 45267, USA; 5Departments of Pediatrics and Molecular & Developmental Biology, Cincinnati Children’s Research Center, Cincinnati, OH 45229, USA

**Keywords:** glutathione, ascorbic acid, oxidative stress, epilepsy, glutamatergic neurotransmission

## Abstract

Reduced glutathione (GSH) and ascorbic acid (AA) are the two most abundant low-molecular-weight antioxidants in mammalian tissues. *Gclm^KO^* knockout mice lack the gene encoding the modifier subunit of the rate-limiting enzyme in GSH biosynthesis; *Gclm^KO^* mice exhibit 10–40% of normal tissue GSH levels and show no overt phenotype. *Gulo^KO^* knockout mice, lacking a functional *Gulo* gene encoding L-gulono-γ-lactone oxidase, cannot synthesize AA and depend on dietary ascorbic acid for survival. To elucidate functional crosstalk between GSH and AA in vivo, we generated the *Gclm^KO^*/*Gulo^KO^* double-knockout (DKO) mouse. DKO mice exhibited spontaneous epileptic seizures, proceeding to death between postnatal day (PND)14 and PND23. Histologically, DKO mice displayed neuronal loss and glial proliferation in the neocortex and hippocampus. Epileptic seizures and brain pathology in young DKO mice could be prevented with AA supplementation in drinking water (1 g/L). Remarkably, in AA-rescued adult DKO mice, the removal of AA supplementation for 2–3 weeks resulted in similar, but more severe, neocortex and hippocampal pathology and seizures, with death occurring between 12 and 21 days later. These results provide direct evidence for an indispensable, yet underappreciated, role for the interplay between GSH and AA in normal brain function and neuronal health. We speculate that the functional crosstalk between GSH and AA plays an important role in regulating glutamatergic neurotransmission and in protecting against excitotoxicity-induced brain damage.

## 1. Introduction

Reactive oxygen radicals, derived from molecular oxygen, are often products of normal cellular metabolism. These molecules, commonly referred to as reactive oxygen species (ROS), are highly reactive to induce oxidation-reduction (redox) reactions. Under physiological conditions, low or moderate concentrations of ROS are recognized as playing important regulatory roles in a number of cellular signaling pathways and the induction of mitogenic responses [1,2]. On the other hand, excessive ROS can cause damage to intracellular nucleic acids, lipids, and proteins, thereby initiating pathological processes [1,2]. These harmful effects of ROS leading to potential biological damage typically arise under conditions of oxidative stress, wherein cellular redox homeostasis is disrupted due to an overproduction of ROS, a deficiency in antioxidants, or both [3,4]. The brain generates large amounts of ROS during rapid cellular metabolism and, therefore, is highly vulnerable to oxidative insults; as such, the antioxidant systems in brain cells are essential in protecting against oxidative stress and in maintaining functional integrity of the brain [5,6].

Reduced glutathione (GSH) and ascorbic acid (AA) are the two most abundant low-molecular-weight antioxidants in the brain. GSH is a ubiquitous tripeptide, synthesized in all cell types. The rate-limiting enzyme in GSH de novo biosynthesis is glutamate-cysteine ligase (GCL), which is a heterodimer comprising a catalytic (GCLC) and a modifier (GCLM) subunit [7]. GSH levels in mammalian nerve cells range between ~2 and 4 mM and are slightly higher in glial cells than in neurons [5,6]. Because of its abundance, GSH acts as a scavenger of free radicals and a cofactor for other antioxidant enzymes [6,7]. AA is synthesized in most mammals from glucose exclusively in liver [8]. Extrahepatic tissues obtain AA from plasma by means of sodium-dependent vitamin C transporters SLC23A1 and SLC23A2 [8,9]. However, humans must obtain AA from dietary sources due to nonsynonymous mutations in the normal L-gulonolactone oxidase gene, rendering it a pseudogene (GULOP); consequently, these amino acid alterations cause GULO, the rate-limiting enzyme in AA biosynthesis, to be inactive in humans [10]. The human GULOP pseudogene is orthologous to the mouse Gulo functional gene. AA is most abundant in the brain, and its level is under strict control, such that 30% of normal levels must be maintained—even under conditions of severe AA deficiency [11,12]. The cellular concentration of AA in neurons (~10 mM) is roughly10-fold higher than that in astrocytes (~1 mM). Like GSH, the biological functions of AA are attributed to its reducing properties, including scavenging free radicals and serving as enzyme co-factors in biosynthesis [13,14,15]. Notably, in addition to their antioxidant functions, both GSH and AA have been implicated to act as endogenous neuromodulators in regulating glutamatergic neurotransmission, by which means they may be involved in the inhibition of neuronal excitability and, therefore, neuroprotection [16,17,18,19,20,21,22].

An interplay between GSH and AA has been proposed based on the following observations: (i) GSH can reduce dehydroascorbate to regenerate AA [23,24]; (ii) GSH depletion and its associated pathology in rodent model systems can be prevented by AA treatment [25,26]; and (iii) there is a compensatory increase in either one of them, following depletion of the other [24,27]. To date, little information exists for GSH-AA interactions in the brain, particularly in the intact animal. In this regard, genetic mouse models of GSH deficiency (*Gclm^KO^* mice) [28] and AA deficiency (*Gulo^KO^* mice) [29], respectively, have been developed to investigate the in vivo physiological/toxicological roles of these antioxidants. *Gclm^KO^* mice have 10–40% of normal GSH levels in all tissues examined, yet they show no overt phenotype when unchallenged [28,30]. *Gulo^KO^* mice, like humans, require dietary AA for survival [29]. When deprived of AA supplementation for 5 weeks, *Gulo^KO^* mice show symptoms of clinical AA deficiency, commonly referred to as scurvy [29]. We have previously investigated neurobehavioral phenotypes in GSH deficient *Gclm^KO^* mice [31] and AA deficient *Gulo^KO^* mice [32]. These studies revealed that the depletion of a single antioxidant is associated with rather subtle neurological changes. To test the hypothesis that the existing antioxidant in the single-knockout mouse provides compensatory neuroprotection, the main aim of the current study is to evaluate phenotypic outcome(s) of depleting both antioxidants in the intact animal. Therefore, we generated a *Gclm^KO^/Gulo^KO^* double-knockout (DKO) mouse model by intercrossing the two single-knockout strains. The resultant DKO mice showed a dramatic neurological phenotype—manifested as spontaneous generalized seizures and premature death.

## 2. Materials and Methods

### 2.1. Chemicals and Reagents

All chemicals and reagents were purchased from Sigma-Aldrich (St. Louis, MO, USA) unless otherwise specified.

### 2.2. Animals

The *Gclm^KO^* mouse line was previously characterized [28] and has been backcrossed to C57BL/6J (B6) background for more than 10 generations. The *Gulo^KO^* mouse line was previously described [29], and its heterozygous breeders, also on the B6 background, were purchased from the Mutant Mouse Regional Resource Centers (MMRRC, http://www.mmrrc.org, accessed on 1 March 2005). Mice were housed in a temperature-controlled room (21–22 °C) on a 12-h light/dark cycle and supplied with a standard rodent diet (#7012, Harlan Teklad) and water ad libitum, unless specified otherwise. Animals were treated humanely, and all animal experiments were approved by and conducted in compliance with the Institutional Animal Care and Use Committees (IACUC) of the University of Cincinnati (Protocol #05-08-11-01).

### 2.3. Generation of GclmKO/GuloKO DKO Mice and Ascorbic Acid Supplementation

We generated the DKO mice by intercrossing *Gclm^KO^* mice with *Gulo^KO^* mice. *Gclm^KO^* females are subfertile [33]; therefore, the breeding scheme was designed to avoid any issue with fertility (Figure 1A). *Gulo^KO^* female and male breeders were provided with *L*-ascorbic acid (AA) in the drinking water (1 g/L) at all times to keep them healthy and fertile. The genotypes of the offspring were determined by PCR analysis of genomic DNA extracted from a 2-mm ear punch collected at postnatal day (PND) 12; PCR primers and protocols used for the detection of *Gclm* and *Gulo* alleles, respectively, have been previously described [28,29]. All pups were weaned between postnatal day (PND) 21 and 28. Body weights were recorded every other day starting at PND12 till PND22. For AA rescue experiments, AA supplementation to pre-weanling mice started at PND16 by providing small bottles of AA-containing drinking water (1 g/L) inside the nursing cages; at this time, pups had begun to walk around and could drink water voluntarily. Surviving post-weanling DKO mice were kept on AA-containing drinking water (1 g/L) until specified ages—when AA was removed from the drinking water or until the study was concluded. Freshly prepared AA-containing water was supplied twice weekly.

### 2.4. Electroencephalography (EEG) Recording

EEG live recordings were performed on three to four mice from each genotype group at the age of PND18-20 as described [34,35]. Briefly, on the day of recording, mice were anesthetized with 3.5% isoflurane in oxygen for the implantation of electrodes (Plastics One Inc., Roanoke, VA, USA) and the connection to a 32-channel EEG machine (Cadwell Laboratories, Kennewick, WA, USA). Mice were allowed to recover from anesthesia and move freely before recording. EEGs were recorded in awake and unrestrained mice for 360 min continuously, after which mice were euthanized by carbon monoxide asphyxiation. EEG data were digitally recorded and analyzed using the Easy EEG II software V. 1.5 (Cadwell Laboratories, Kennewick, WA, USA). Electrographic seizures were defined as the sudden onset of high amplitude (>2 × background) activity with signal progression (a change in amplitude and frequency over the course of the event) and a duration greater than ten seconds. EEG records were inspected by a clinically-qualified electroencephalographer (KDH) blinded to genotype groups.

### 2.5. Histological and Immunohistochemical Analyses

Mice were anesthetized with an intraperitoneal injection of avertin then perfused transcardially with freshly made 4% paraformaldehyde. Brains were excised and post-fixed in 4% paraformaldehyde at 4 °C overnight and then sent to the Department of Comparative Pathology at the University of Cincinnati for the processing of paraffin embedding, preparation of coronal sections (5 µm thick) through the dorsal hippocampus, and staining of brain coronal sections by hematoxylin and eosin (HE) and toluidine blue O (TBO) following standard procedures. Terminal deoxynucleotidyl-transferase-mediated dUTP-biotin nick-end-labeling (TUNEL) staining was performed on paraffin sections using In Situ Cell Death Detection kit (Roche; Indianapolis, IN, USA) according to the manufacturer’s protocol. Immunohistochemical (IHC) staining for glial fibrillary acidic protein (GFAP) was conducted on paraffin sections using TSA Biotin System Kit (NEN Life Science Products, Boston, MA, USA), according to the manufacturer’s protocol. The primary antibody against mouse GFAP (ThermoFisher, Waltham, MA, USA) was used at a dilution of 1:100, and horseradish peroxidase (HRP)-conjugated secondary antibody (ThermoFisher, Waltham, MA, USA) was used at a dilution of 1:200. Images were obtained using a Nikon Eclipse TE-300 microscope.

### 2.6. Biochemical Measurements of Redox Molecules

Mice were euthanized by carbon dioxide asphyxiation and decapitated. Neocortex, hippocampus, and cerebellum regions were carefully dissected out, snap-frozen in liquid nitrogen, and stored at −80 °C for later biochemical assays. At the time of assay, frozen brain tissues were homogenized in ice-cold redox-quenching buffer (20 mM HCl, 5 mM diethylenetriaminepentaacetic acid, 5% trichloroacetic acid), followed by centrifugation at 12,000× *g* for 10 min. The resulting supernatants were processed for the determination of GSH and glutathione disulfide (GSSG, oxidized glutathione) levels using the fluorescent probe *o*-phthalaldehyde as described [36] and of AA levels by spectrophotometric measurement of ferritin iron released by AA, as previously described [37]. Results are expressed as μmol/g tissue weight. The malondialdehyde (MDA) levels were measured using the Bioxytech Lpo 586 Kit (Oxis International, Inc., Foster City, CA, USA) according to the manufacturer’s protocol. Protein concentrations of tissue homogenates were measured using the BioRad protein-assay reagents (BioRad, Hercules, CA, USA), according to the manufacturer’s protocol. MDA levels are expressed as nmol/mg protein.

### 2.7. Measurements of Hematocrit and Plasma Glucose and Lipid Profile

Mice were euthanized by carbon dioxide asphyxiation and blood was collected by cardiac puncture. Droplets of blood were collected with hematocrit capillary tubes and centrifuged for 2 min, using a hematocrit centrifuge (Fisher Scientific, Waltham, MA, USA). Hematocrit values were manually determined as the percentage of blood composed of red blood cells. Whole blood was centrifuged at 2000× *g* for 5 min at 4 °C, and the supernatant plasma was stored at −80 °C for later metabolite measurements. Plasma glucose, electrolyte levels, and lipid profiles (including triglyceride (TG), phospholipids (PL), cholesterol (Chol), and nonesterified free fatty acids (NEFA)) were measured by the Mouse Metabolic Phenotyping Center at the University of Cincinnati Medical Center, according to established protocols.

### 2.8. Statistics

Group differences were analyzed using Graphpad Prism software (San Diego, CA, USA). Differences between two groups were analyzed by Student’s unpaired t-test. Multiple-group comparisons were performed by one-way ANOVA. A post-hoc Bonferroni test was performed to accommodate multiple comparisons. Results are presented as means ± S.D. A score of *p* < 0.05 was considered statistically significant.

## 3. Results

### 3.1. Concomitant Deficiency in GSH and AA Biosynthesis Leads to Growth Retardation, Spontaneous Seizures, and Premature Death

Both *Gclm^KO^* and *Gulo^KO^* mouse strains have been maintained on the C57BL/6J background. The two KO mouse strains were intercrossed to generate DKO mice following the breeding scheme shown in Figure 1A. Given that one allele of either the *Gclm* [28] or *Gulo* [29] gene is sufficient to maintain normal tissue GSH and AA levels, respectively, the breeding scheme was designed to simplify the breeding process and used the compound heterozygote as our control, when comparing the homozygous knockout of either gene alone, or both genes together. Offspring of *Gclm^HET^*/*Gulo^HET^* (HET), *Gclm^KO^/Gulo^HET^* (MKO), *Gclm^HET^/Gulo^KO^* (GKO), and *Gclm^KO^/Gulo^KO^* (DKO) genotypes (Figure 1B and Appendix A) were born in the expected Mendelian ratios, indicating that the double ablation of *Gclm* and *Gulo* genes did not cause embryonic lethality. By weaning time—i.e., around PND28—the HET, MKO, and GKO mice all displayed similar growth curves (Figure 1C); however, DKO mice of both genders showed growth retardation (Figure 1C) and increased mortality (Figure 1D), with the first death observed at PND14 and zero survival beyond PND23.

Within 24 h preceding death, by visual observation, DKO mice exhibited seizure behavior—including ataxia, wild running, freezing posture, straub tail, and tonic-clonic generalized seizures; in some mice, status epilepticus leading to immediate death was noted. In agreement with epileptic seizure behavior, spontaneous electrographic seizures were detected in all examined DKO mice (*N* = 4) by EEG at PND18 to PND20 (Figure 1E). At the same ages, HET, MKO, and GKO mice showed neither seizure behavior nor electrographic seizures in all mice examined (*N* = 3/genotype; Figure 1E). Considering the global disruption of *Gclm* and *Gulo* genes in these mice, to exclude the possibility that seizure activity was induced by metabolic changes, we examined plasma cations (Na^+^, K^+^, Ca^++^), glucose levels, and lipid profiles (Table 1). No abnormalities and no differences were found among the four genotype groups. Measurements of hematocrit showed that GKO and DKO mice did not develop clinical AA deficiency (e.g., scurvy) at this age (Table 1); this finding had been expected because these pre-weanling mice were nursed by dams supplied with AA (1 g/L) in the drinking water. Based on all these findings, and the link between epileptic seizures and death due to persistent apnea and/or airway obstruction, we concluded that the epileptic seizure was likely to be the cause of premature deaths seen in DKO mice.

### 3.2. DKO Mice Show Neuronal Loss and Glial Proliferation in the Neocortex and Hippocampus

The epileptic seizure phenotype of DKO mice prompted us to examine the histomorphology of the brain. At PND18-PND20, no gross morphological changes were found. The brain mass, as a percentage of body weight, was 5.5 ± 0.3, 5.9 ± 0.5, 5.5 ± 0.4, and 6.0 ± 0.3 in HET, MKO, GKO (*N* = 7/genotype), and DKO mice (*N* = 9), respectively (*p* > 0.05 by one-way ANOVA). HE staining of brain coronal sections (Figure 2A) revealed no apparent abnormalities, except that the neocortex of DKO mice appeared thinner, when compared to that of other genotypes. However, TBO staining (Figure 2B) revealed neuronal loss and reactive glial-cell proliferation in the neocortex (predominantly in the II/III layers) and in the hippocampal dentate gyrus/*cornu ammonis*-3 and hilus (DG/CA3-H) fields of DKO mice (Figure 2B, *red arrow heads*). TUNEL-positive cells (Figure 2C)—indicating apoptotic cells—were sporadic in the neocortex of DKO mice (Figure 2C, *black arrow heads*), but they were more predominant in the CA3 (Figure 2C, *red arrows*) and DG/H regions (Figure 2C, *blue arrows*) as well as in the DG granule-cell layer (Figure 2C, *black arrows*). IHC staining for GFAP, a marker for astroglial cells, confirmed the reactive glial proliferation in these areas with neuronal death (Figure 2D). It should be noted that this brain pathology was observed in DKO mice, which both did and did not exhibit epileptic behavior at the time of sacrifice. We did not find similar lesions in the hippocampus proper or other brain regions of the DKO mice, nor in any brain regions of HET, MKO, or GKO mice.

### 3.3. Brains of Young DKO Mice Show Region-Specific GSH and AA Deficiencies and Lipid Peroxidation

The above observations indicated that epilepsy and associated pathology occurred only when GSH and AA biosynthesis were both compromised. In providing GSH and AA to extrahepatic tissues, the liver is the primary organ of GSH de novo biosynthesis and the only source of circulating AA in the absence of dietary supplementation [8,28]. As such, we measured GSH and AA levels in the liver and brain at PND18 (Figure 3). It should be kept in mind that, at this time, these mice were still being breast-fed by dams supplemented with AA. When comparing HET to GKO mice or MKO to DKO mice, liver GSH levels were comparable (Figure 3A), indicating that the absence of either one or both *Gulo* functional alleles did not affect hepatic GSH concentrations. Similarly, liver AA levels were no different between HET and MKO mice or between GKO and DKO mice, indicating that absence of either one or both *Gclm* functional alleles did not alter hepatic AA concentrations (Figure 3A).

In the brain (Figure 3B–E), GSH and AA were measured in neocortex and hippocampus, the two regions that showed neuronal damage in epileptic DKO mice; the cerebellum was examined as a control region, in which no pathology had been observed. For GSH (Figure 3B), MKO and DKO had comparable low levels (~25–35% of HET) in all three regions; interestingly, GSH from GKO mice also declined to ~74% in neocortex (*p* = 0.0002), ~78% in hippocampus (*p* = 0.007), and ~80% in cerebellum (*p* = 0.056) of HET levels. For GSSG levels (Figure 3C), MKO and DKO mice had ~35–50% of HET levels in neocortex and cerebellum regions; however, in the hippocampus of GKO and DKO mice, GSSG was increased to ~1.8- to 2.0-fold of HET levels. Such changes in GSH versus GSSG (Figure 3D) led to significant declines in the GSH/GSSG ratios (an index of the reduction status of total glutathione pool) in neocortex and hippocampus of GKO and DKO mice, albeit DKO hippocampus was affected more dramatically. For AA levels (Figure 3E), MKO mice were no different from HET mice in any of the three regions, and GKO mice had ~40–46% of HET levels in these brain tissues. However, AA was profoundly depleted (<25% of HET levels) in the DKO neocortex and hippocampus. Oxidative stress in these brain regions were assessed by measuring MDA, the by-product of lipid peroxidation (Figure 3F); MDA levels were found to be ~1.6-fold of HET levels in neocortex and hippocampus of DKO mice. Lastly, we examined the gene expression of GSH-metabolizing enzymes (*Gcl*, *Gsr,* and *Gpx*) and AA transporters (*Slc23a1* and *Slc23a2*) in neocortex and hippocampus (Appendix A). Intriguingly, no alterations in these genes were observed despite of the dramatic redox imbalance in these brain regions of DKO mice.

### 3.4. AA Rescues DKO Mice, but Removal of AA in Rescued DKO Mice Results in Spontaneous Seizures and Premature Death

Our biochemical assays show that the major effect—caused by loss of both *Gclm* and *Gulo* genes—is profound AA depletion in specific brain regions of the DKO mice at PND18. Therefore, we attempted to rescue pre-weanling DKO mice with AA supplementation. Because all DKO mice died by PND23 even when the nursing females were fed AA-supplemented water, small bottles of freshly made water containing AA (1 g/L) were provided in nursing cages starting at PND16; this is when pups start to drink water voluntarily on their own. This regimen was successful in rescuing DKO mice that had not shown any observable seizure activities by PND16; however, it was only able to rescue one third of DKO mice that had already exhibited seizures.

Rescued DKO mice had lower body weights when compared to age- and gender-matched HET mice at ages 1 month and 3 months (Figure 4A). One pilot rescue experiment lasted till the age of 6 months, at which time two male DKO mice grew up to comparable body weights of HET mice. Furthermore, no epileptic behavior was visually observed in these rescued DKO mice. However, when we stopped AA supplementation at the age of 1 or 3 months, DKO mice appeared normal by 13 days without AA, after which time they showed a trending loss of body weight (Figure 4B) and increased mortality with zero survival beyond 22 days on AA removal (Figure 4C). DKO mice on AA removal appeared less active; tonic-clonic generalized seizures were only seen in one out of eleven mice and many deaths occurred unexpectedly. EEG analysis, performed on a few adult DKO mice at 14–17 days after AA removal, revealed multiple episodes of electrographic seizures with high frequency.

Brains collected at 17–20 days after AA removal from adult epileptic DKO mice showed multiple types of pathology (Figure 5). In all younger (e.g., 1-mo-old; *N* = 5) and greater than half the older (e.g., 3-mo-old; *N* = 3 out of 5 total), DKO mice that were examined, gross morphology of the brain was preserved (Figure 5A). However, TBO staining (Figure 5B) revealed extensive neuronal degeneration in the neocortex and the CA1-3 pyramidal-cell layers and DG granule-cell layers in the hippocampus. TUNEL-positive cells, indicating cell death (Figure 5C), and GFAP-immunopositive cells denoting glial proliferation (Figure 5D) were striking in these regions. On the other hand, two older DKO mice that we examined exhibited severe morphological abnormalities in brain, including atrophy of the neocortex, dilated lateral ventricles, and substantial damage in the hippocampus (Figure 5A). The above noted pathological changes were not observed in the brains of HET mice or DKO mice kept on AA supplementation (Figure 5A). Moreover, the same panel of plasma metabolites were measured in adult DKO mice (Table 1). Compared to those kept on AA supplementation, DKO mice with AA removal for 17–20 days had lower glucose and higher cholesterol levels in plasma. Similarly, as seen in young DKO mice, adult epileptic DKO mice had normal hematocrit levels, suggesting that AA removal for up to 20 days did not cause clinical AA deficiency (e.g., scurvy) in these adult mice.

## 4. Discussion

Cellular antioxidant systems are redundant and diverse. These antioxidants cooperate to maintain the overall cellular redox balance on one hand, and they have specific preferences for substrates and subcellular locations, on the other hand [38,39,40]. It is therefore likely that interactions among antioxidants might be physiologically important in different tissues. Herein, we report, for the first time, that the combined deficiency of GSH and ascorbate in mice is a direct cause of epileptic seizures leading to premature death—not only in pre-weanling pups, but also in adult rescued mice when dietary AA was removed. This dramatic neurological phenotype, observed in DKO (*Gclm^KO^*/*Gulo^KO^*) mice, implies a fundamental and yet underscored role of the interplay between GSH and AA in normal brain function and neuronal health. At the redox biochemistry level, our data show that: (i) GSH or AA supplied by the mother’s breast feeding was not sufficient to sustain normal levels of these antioxidants in the brain of the offspring, when de novo biosynthesis in the offspring was compromised; (ii) GSH concentration and/or oxidation were altered by AA deficiency in a region-specific way; and (iii) AA depletion was worsened by GSH deficiency in the neocortex and hippocampus, where lipid peroxidation was greatly enhanced.

Epileptic seizures stem from abnormal electrical activities in the brain, commonly associated with disruption of synaptic excitation/inhibition balance—leading to neuronal hyperexcitability [41,42]. The contribution of an excitatory glutamatergic mechanism to epileptogenesis is supported by numerous lines of evidence [43,44,45,46]. In this context, both GSH and AA have been implicated in modulating glutamatergic neurotransmission. First, GSH may modulate synaptic transmission and plasticity via redox-sensitive proteins, including *N*-methyl-D-aspartate (NMDA) receptors, calcium channels, and glutamate transporters [16,17,18,20]. Second, GSH has been proposed to bind to glutamate receptors (e.g., NMDA receptors) via its γ-glutamyl moiety and to displace glutamate agonists [47,48,49]. Third, AA has been suggested to regulate glutamatergic neurotransmission through glutamate-ascorbate hetero-exchange at glutamatergic synapses [21,50]. It is proposed that micromolar increases in AA release might facilitate uptake of intersynaptic glutamate; at millimolar concentrations, AA is known to display inhibitory effects on NMDA receptors [19]. Thus, the fine-tuning effect by GSH and AA on glutamatergic neurotransmission agrees with a neuroprotective role through controlling neuronal excitability. We speculate that loss of this coordinated regulation, such as that seen in DKO mice, may promote seizure activities.

It is well documented that oxidative stress is an important consequence of glutamate receptor activation and excitotoxicity in experimental epileptic models—as indicated by common sequelae of ROS formation including lipid peroxidation [51,52] and oxidative DNA damage [53,54]. An increase in ROS formation seems to be a major molecular event preceding neuronal cell death after seizure attack [55,56]. In this regard, the antioxidant functions of GSH and AA are pivotal in maintaining neuronal redox homeostasis at physiological levels of excitatory neurotransmission, as well as in protecting against oxidative brain damage following seizure activities. In the current study, either in pre-weanling or in adult epileptic DKO mice, we observed neuronal death and reactive glial proliferation in neocortex and more severe in hippocampus, the two brain areas most commonly involved in generating seizures [57]. These types of brain pathology likely reflect a consequence of excessive excitotoxicity, arising from seizure episodes in DKO mice. Furthermore, redox biochemistry analyses revealed the presence of enhanced oxidative stress in these brain tissues of pre-weanling DKO mice, as evidenced by the oxidation of already-depleted GSH pools and elevated lipid peroxidation. However, it is unknown whether oxidative stress—intrinsic to GSH and AA deficiency in the DKO brain—was causally involved in seizure development, aside from being an early cellular change resulting from seizure activities. In addition, in our mouse model, we have not examined the status of other antioxidants that have been implicated in altering susceptibility to seizure, such as vitamin E [58], uric acid [59], and L-carnitine [60]. Thus, the possibility that changes in one or more of other antioxidant molecules may contribute to the epilepsy phenotype of DKO mice cannot be excluded. Future studies with a careful characterization of the temporal progression of redox biochemistry, glutamatergic neurotransmission, seizure activities, and brain pathology—as well as antioxidant intervention studies—in these mice are warranted to define the exact cause-and-effect relationship.

Epilepsy is a common neurological disorder that can last a lifetime. It is a major public health problem due to the significant consequences and costs incurred in afflicted individuals and society. Despite these challenges, our understanding of the molecular mechanisms underlying epileptogenesis remains limited. To date, more than 100 epilepsy mouse models have been reported. Data obtained from these studies [61,62,63] confirm the essential role of perturbed synaptic function in seizure development; more importantly, they suggest the complexity of this process, which begs for the need of a more precise understanding of the regulatory mechanisms of synaptic neurotransmission. The fatal-seizure phenotype associated with compromised biosynthesis in both GSH and AA in mice highlights an important function of the crosstalk between these two molecules in the brain. We hypothesize such crosstalk has two layers (Figure 6): first, both molecules may act as neuromodulators that coordinately regulate excitatory glutamatergic neurotransmission; second, through mutual compensatory effects, they are antioxidants responsible for maintaining cellular redox homeostasis and for protecting neuronal cells against oxidative injury. Thus, although not intended to exemplify the human situation, the DKO mouse strain nevertheless represents an intriguing model system that could elucidate novel redox mechanisms underlying regulation of synaptic neurotransmission and associated neurological disorders.

Lastly, the current study has implications for human neuronal health. It is well established that GSH deficiency is intimately involved in the pathogenesis of a variety of neuronal diseases [6,64]. Individuals carrying functional mutations in GSH biosynthesis genes are reported to manifest progressive neurological degeneration—including mental retardation, ataxia, and seizures [65]. In patients with epilepsy, a widespread impairment of the GSH biosynthetic pathway, independent of seizure activity, has been reported [66]. On the other hand, although the clinical deficiency of AA leading to scurvy is rare in modern days, recent reports highlight the underestimated high prevalence of subclinical AA deficiency worldwide, including the U.S., particularly in low- and middle-income communities [67,68,69]. Furthermore, epidemiological studies show that low AA is common during pregnancy, accounting for up to 25–30% of parturient women [70,71]; the status of low AA may be worsened in individuals who are heavy smokers and/or consume excessive alcohol [70,71], both of which habits are known to cause systemic decreases in GSH levels [72,73]. Thus, the potential co-occurrence of GSH and AA deficiencies in susceptible populations, and particularly during pregnancy, may have unrecognized detrimental impacts on the brain of the developing embryo and fetus.

## 5. Conclusions

In summary, we report a fatal epilepsy phenotype in mice that is caused by the dual deficiency in de novo biosynthesis of GSH and AA, the two major endogenous antioxidants. This severe neurological phenotype is accompanied by excessive oxidative stress and neuronal death noted predominantly in the neocortex and hippocampus of affected animals, supporting the necessity of GSH and AA interplay in the maintenance of redox homeostasis and neuronal integrity in these brain regions. The aforementioned mouse model holds promise in elucidating novel redox-associated mechanisms underlying epileptogenesis, such as redox regulation of glutamatergic neurotransmission and cellular defense against excitotoxicity.

## Figures and Tables

**Figure 1 antioxidants-12-00448-f001:**
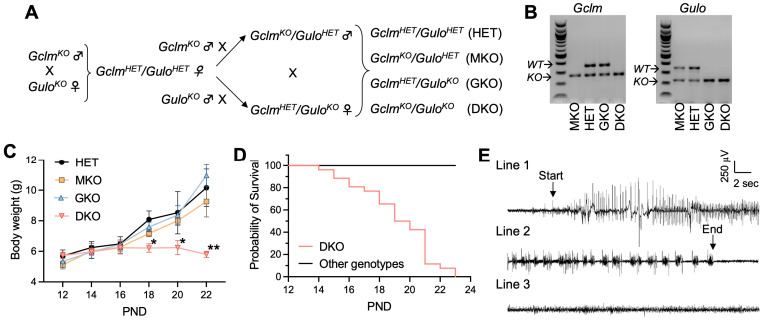
Genetic deficiency in GSH and AA biosynthesis leads to growth retardation, spontaneous generalized seizure and premature death. (**A**), Breeding scheme—to obtain HET, MKO, GKO and DKO lines. (**B**), Representative electrophoresis images of PCR genotyping of offspring for *Gclm* and *Gulo* alleles. PCR products derived from wild-type (WT) or knockout (KO) alleles of respective genes are labeled. (**C**), Body weight as a function of postnatal day (PND). *N* = 7–8/genotype. Data are expressed as means ± S.D. * *p* < 0.05, ** *p* < 0.01, when compared to age-matched HET mice by Student’s unpaired *t*-test. (**D**), survival rate of HET, MKO, GKO (*N* = 23–25/genotype) offspring vs DKO (*N* = 26) offspring. First death occurred at PND14, and last death happened on PND23. (**E**), Representative EEG of a spontaneous seizure recorded in a DKO mouse at age PND19. The EEG pattern of the onset (Line 1) and the termination of a 90-s seizure (Line 2) are compared with that of a control HET mouse (Line 3).

**Figure 2 antioxidants-12-00448-f002:**
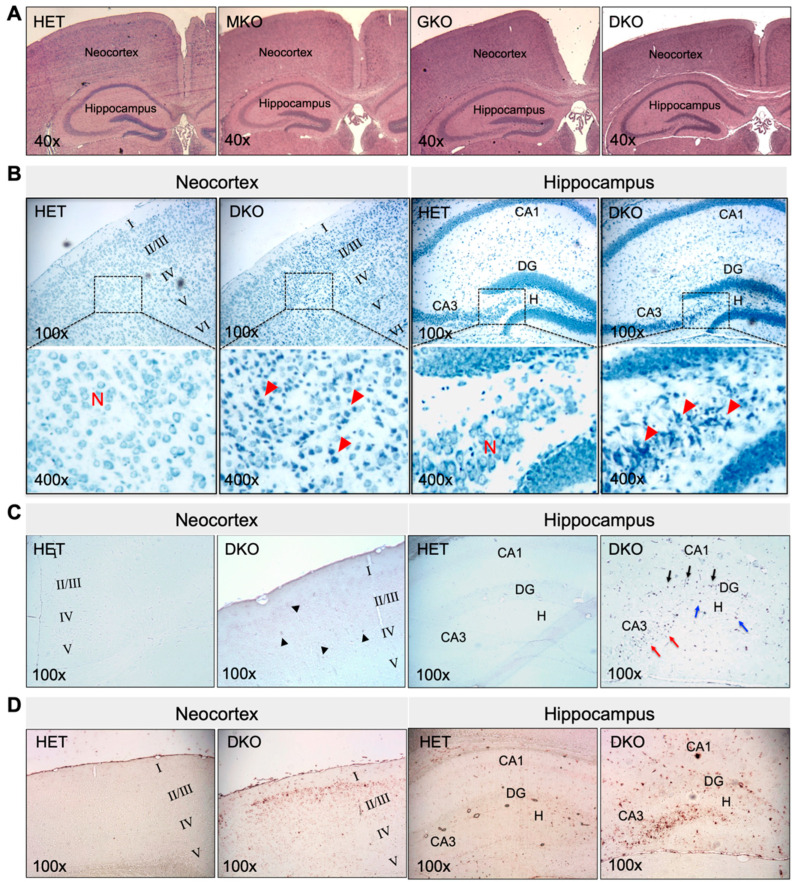
Young DKO mice show neuronal death and glial proliferation in the neocortex and hippocampus at PND18-20. (**A**), Representative images of HE staining of paraffin-embedded brain coronal sections from HET, MKO, GKO and DKO mice. (**B**), Representative images of TBO staining of paraffin-embedded brain coronal sections from HET and DKO mice. The *square area* in the *top panel* was enlarged in the *lower panel*. Loss of neurons (*N*) and reactive glial proliferation (*red arrow heads*) were noted in the neocortex and the hippocampal DG/CA3-H fields of DKO brains. (**C**), Representative images of TUNEL staining of paraffin-embedded brain coronal sections from HET and DKO mice. TUNEL-positive (apoptotic) cells were sporadic in the neocortex (*black arrow heads*) and more predominant in the CA3 (*red arrows*) and DG/H regions (*blue arrows*) and in the DG granule-cell layer (*black arrows*) of DKO brains. (**D**), Representative images of IHC detection of *GFAP* (a marker for astroglial cells) in the neocortex and the hippocampus of HET and DKO brains. HE, hematoxylin and eosin; *N*, neurons; TBO, toluidine blue O; DG, dentate gyrus; CA3-H, cornu ammonis-3; H, hilus; TUNEL, terminal deoxynucleotidyl-transferase-mediated dUTP nick-end-labeling; IHC, immunohistochemistry; GFAP, *α*-glial fibrillary acid protein. Magnifications: 40×, 100×, and 400× as indicated.

**Figure 3 antioxidants-12-00448-f003:**
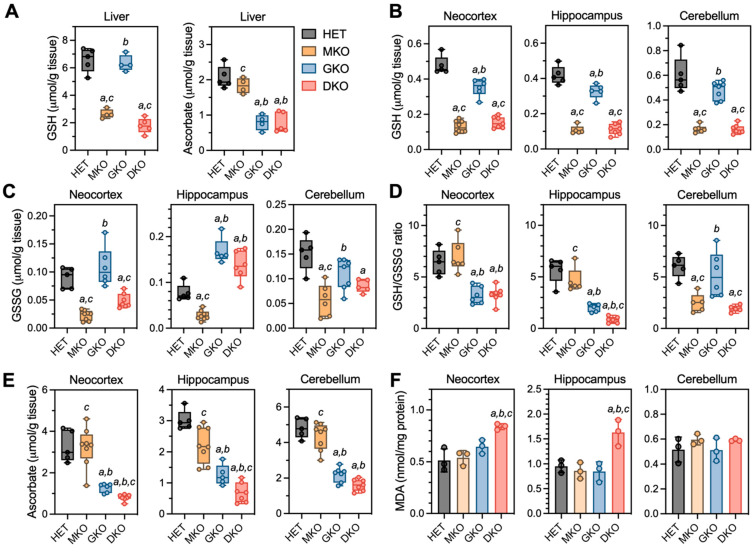
Redox biochemistry in liver and brain regions of young HET, MKO, GKO and DKO mice at PND18. (**A**), Liver tissues were processed for measurements of GSH and AA. Three brain regions were processed for measurements of (**B**), reduced GSH; (**C**), GSSG (oxidized GSH); (**D**), GSH/GSSG ratios (calculated from GSH and GSSG values of each individual animal); (**E**), AA; and (**F**), MDA (malondialdehyde is one of the final products of intracellular polyunsaturated fatty acids peroxidation). Data are expressed as means ± S.D. (*N* = 4–5/genotype for livers and *N* = 5–8/genotype/region for brains). Group differences were analyzed by one-way ANOVA followed by Bonferroni *post-hoc* test. *^a^ p* < 0.05, when compared to HET mice; *^b^ p* < 0.05, when compared with MKO mice; *^c^ p* < 0.05, when compared with GKO mice.

**Figure 4 antioxidants-12-00448-f004:**
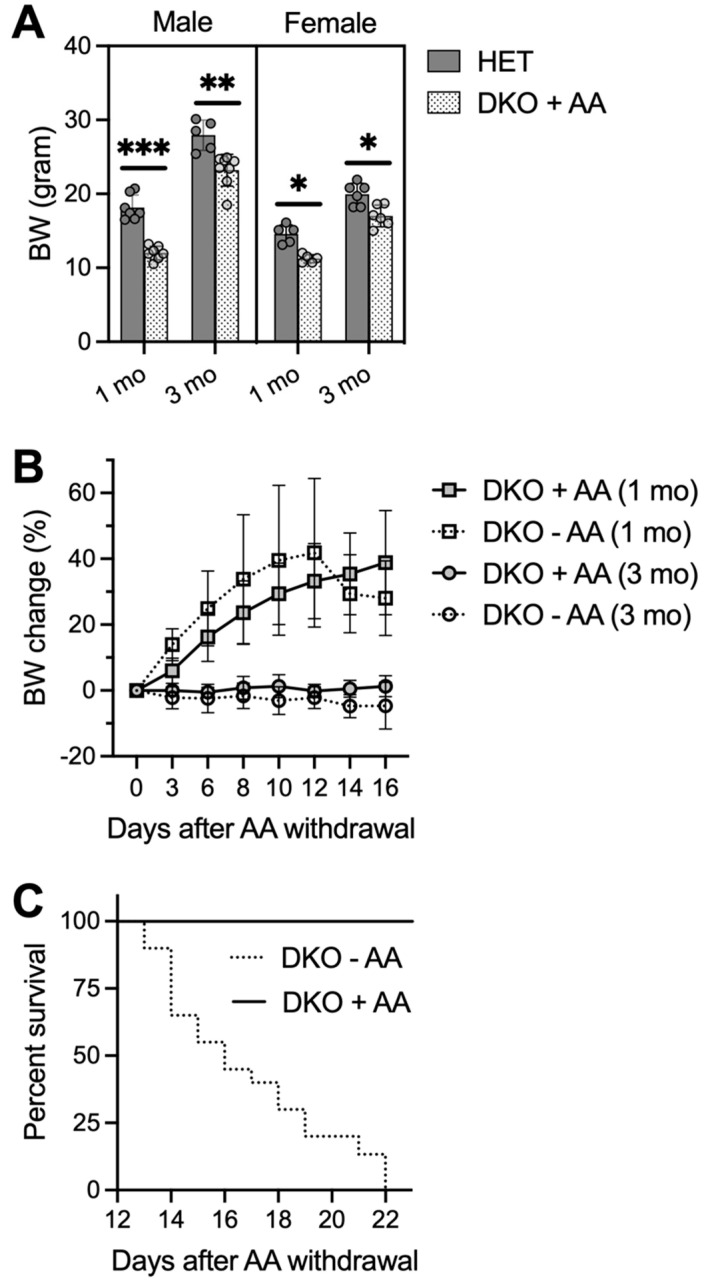
AA-rescued DKO mice die 2-3 weeks after AA removal. Small bottles of freshly-prepared water containing AA (1 g/L) were provided for pre-weanling pups in nursing cages starting at PND16. AA-rescued DKO mice were weaned at PND21-23 and kept on AA-supplemented drinking water (1 g/L) until at the age of 1 or 3 months when AA was removed from the drinking water or until conclusion of the study. (**A**), body weights (BW) of male and female HET and AA-rescued DKO mice at ages of 1 and 3 months (*N* = 6–8/group). (**B**), BW change of AA-rescued DKO mice after AA removal (−AA) or maintained on AA (+AA) (*N* = 5–6/group). (**C**), survival rate of AA-rescued DKO mice after AA removal (−AA; *N* = 19) or maintained on AA (+AA; *N* = 15). Data are expressed as means ± S.D. * *p* < 0.05, ** *p* < 0.01, *** *p* < 0.001, when compared to age- and gender-matched HET mice by Student’s unpaired *t*-test.

**Figure 5 antioxidants-12-00448-f005:**
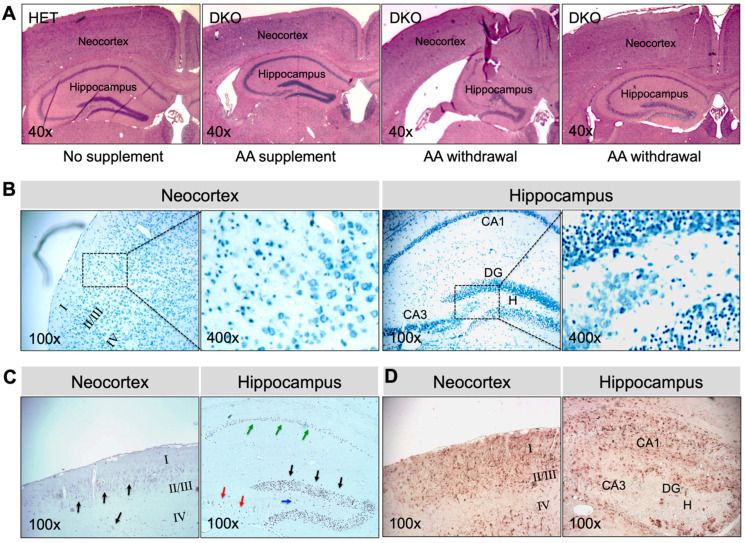
Brain pathology in AA-rescued DKO mice at 17–20 days after AA removal. (**A**), Representative images of HE staining of paraffin-embedded brain coronal sections from HET mice, AA-rescued DKO mice kept on AA supplementation, and AA-rescued DKO mice with AA removal (starting at age 1 month or 3 months) for 17–20 days. Representative images of (**B**), TBO staining, (**C**), TUNEL staining, and (**D**), IHC staining for GFAP of paraffin-embedded brain coronal sections from AA-rescued DKO mice with AA removal (starting at age 1 month) for 20 days. In (**B**), the *square area* in the *left panel* was enlarged in the right panel. In (**C**), *arrows* showing TUNEL-positive (apoptotic) cells. HE, hematoxylin and eosin; TBO, toluidine blue O; DG, dentate gyrus; CA1, cornu ammonis-1; CA3, cornu ammonis-3; H, hilus; TUNEL, terminal deoxynucleotidyl-transferase-mediated dUTP nick-end-labeling; IHC, immunohistochemistry; GFAP, *α*-glial fibrillary acid protein. Magnifications: 40×, 100×, and 400× as indicated.

**Figure 6 antioxidants-12-00448-f006:**
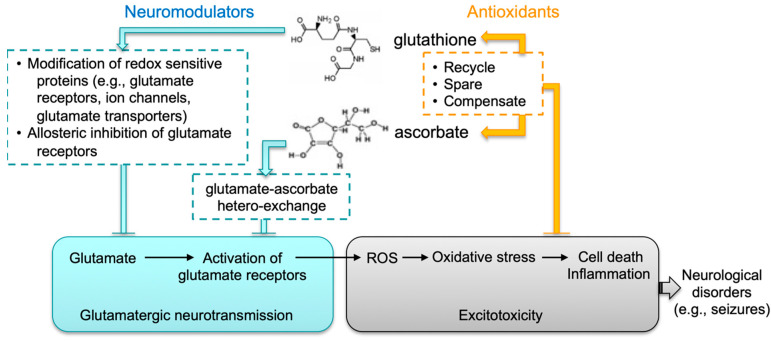
Proposed functional crosstalk between GSH and AA in glutamatergic neurotransmission. We speculate that the interplay between glutathione and ascorbate in brain has two major functions: (1) Both molecules likely act as neuromodulators that coordinately control excitatory glutamatergic neurotransmission. GSH may modulate synaptic transmission and plasticity via redox-sensitive proteins, including glutamate receptors, calcium channels, and glutamate transporters. GSH may bind to glutamate receptors via its γ-glutamyl moiety and displace glutamate agonists. AA may regulate glutamatergic neurotransmission through GSH-AA hetero-exchange at glutamatergic synapses. (2) Through mutual compensatory effects as antioxidants (e.g., recycling, sparing and compensating), GSH and AA are responsible for maintaining cellular redox homeostasis and protecting against excitotoxicity.

**Table 1 antioxidants-12-00448-t001:** Plasma metabolites profile and hematocrit.

Genotype (*N* of Mice)	AA Supplement	Glu ^1^	TG	PL	Chol	NEFA	Hct
		(mg/dL)				(mEq/L)	(%)
Pre-weanling mice (PND17-20)							
*HET (5)*	none	273 ± 21	88 ± 22	192 ± 20	115 ± 8	0.9 ± 0.2	49 ± 2
*MKO (4)*	none	277 ± 4	107 ± 31	207 ± 14	156 ± 11	1.2 ± 0.3	49 ± 2
*GKO (4)*	none	281 ± 10	77 ± 11	197 ± 18	145 ± 17	1.5 ± 0.4	48 ± 2
*DKO (5)*	none	223 ± 14	85 ± 14	195 ± 15	136 ± 18	0.7 ± 0.1	51 ± 3
AA-rescued mice (1 or 3 mo)							
*DKO (8)*	1 g/L	290 ± 36	49 ± 9	198 ± 39	94 ± 22	0.6 ± 0.1	47 ± 1.2
*DKO (9)*	Removal ^2^	159 ± 85 ^†^	85 ± 42	269 ± 89	138 ± 35 ^†^	1.0 ± 0.4	49 ± 2.7

^1^ Glucose levels were taken from non-fasted mice and are not in the diabetic range. ^2^ Surviving post-weanling DKO mice were kept on AA-supplemented drinking water (1 g/L) until AA was removed at the age of 1 or 3 mo. Measurements were taken at 17–20 days after AA removal. Plasma Na^+^, K^+^ and Ca^2+^ levels were determined, and they were all within the normal range. Values are expressed as means ± S.D. ^†^
*p* < 0.05, when compared to aged-matched DKO mice kept on AA-supplemented water. AA, ascorbic acid; Glu, glucose; TG, triglyceride; PL, phospholipids; Chol, cholesterol; NEFA, nonesterified free fatty acids; Hct, hematocrit.

## Data Availability

The data presented in this study are available on request from the corresponding author.

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
