# Peer review of "Fatal Epileptic Seizures in Mice Having Compromised Glutathione and Ascorbic Acid Biosynthesis"

_antioxidants, 2023, doi:10.3390/antiox12020448_

Round 1

Reviewer 1 Report

The overall impression of the contribution of the current study “Fatal Epileptic Seizures in Mice Having Compromised Glutathione and Ascorbic Acid Biosynthesis” is reasonable. However, the Authors may consider doing necessary amendments to the manuscript for better comprehensibility of the study.

It would be interesting to know

1)    the activities of glutathione metabolizing enzymes such as GR, GPx in GclmHET/GuloHET (HET), 201 GclmKO/GuloHET (MKO), GclmHET/GuloKO (GKO) and GclmKO/GuloKO (DKO) mice,

2)    Role/activity of L-ascorbate oxidase, and

3)    Role/activities of Sodium-dependent Vitamin C Transporter (SVCT) 1 and 2

Please cite some general articles (https://pubmed.ncbi.nlm.nih.gov/10569624/, https://www.tandfonline.com/doi/abs/10.1080/10715762.2019.1702656)

where glutathione and ascorbic acid (AA) in different conditions were studied and their role in various conditions pathological conditions.

Author Response

#1: We agree that investigation of these enzymes and transporters of the GSH-AA pathways will provide useful information for a better characterization of our mouse model. Unfortunately, we do not have biological samples available to perform these assays. To try to partially address this comment, with available mRNA samples, we examined the gene expression levels by real-time PCR analysis. We have presented these new data as Supplemental Fig. 1 in this revision. Our results showed that, except the expected changes in Gclm and Gulo genes depending on the genotype group, there were no differences in any examined genes among four genotype groups.

#2: Some general articles have been cited.

Reviewer 2 Report

Review of manuscript ref. number: antioxidants-2180981

Title: Fatal Epileptic Seizures in Mice Having Compromised Glutathione and Ascorbic Acid Biosynthesis

Authors: Ying Chen, Katherine D. Holland, Daniel W. Nebert.

General comment: the study reports the neurological alterations in mice double knockout for reduced glutathione and ascorbic acid. The authors establish the double knockout mice and evaluate some biochemical and histological parameters before the animals develop fatal epileptic seizures. The authors conclude that there is a crosstalk between both major endogenous antioxidants, both as neuromodulators that coordinately regulate excitatory glutamatergic neurotransmission, and through mutual compensatory effects to maintain cellular redox homeostasis. The hypothesis and objectives are sound and clearly exposed, methods seem adequate and results are attractive, well discussed and with a translational potential to humans. Most intriguing issue is the authors’ surprise for the dramatic outcome of the double deprivation of the two main endogenous antioxidants. Some specific comments are detailed below:

Specific comments:

1)      Abstract, first line: GSH usually stands for reduced glutathione, not simply glutathione. It also applies to line 51.

2)      Line 43; it should be nucleic.

3)      Line 85; the fact that DKO mice, deprived of the two major endogenous antioxidants at the moment of maximum brain growth spurt, showed a dramatic neurological phenotype with spontaneous generalized seizures and premature death should not be that surprising.

4)      Lines 100-103; protocol reference number for the animal experiment should be included.

5)      Line 165; GSSG should be spelled out here as oxidized glutathione.

6)      Line 216; same as above, the authors reiterate that the severe neurological phenotype and early mortality in DKO mice had not been expected. In my opinion, since perinatal stage in rodent is the period of maximum growth spurt in brain, highest metabolic rate in neurons and glia and peak ROS generation, deprivation of the two major endogenous antioxidants should dramatically affect brain development and function. Indeed, the increased MDA levels in neocortex and hippocampus depicted in Figure 3F, suggesting a significant damage in membrane lipids of brain cells, could clearly explain the altered neuronal functionality. Even in adulthood, such as in the rescue experiment, brain is still a major consumer of glucose and producer of oxygen species that need to be controlled; hence, seizures are back when AA rescue is discontinued.

7)      Lines 413-417; since the two references quoted to support the connection between oxidative stress and epileptic seizures are from 2 decades ago, perhaps the authors should update the bibliography with a very recent review from this year: Oxidative stress: a common imbalance in diabetes and epilepsy. Ramos-Riera et al. Metab Brain Dis. 2023 Jan 4. doi: 10.1007/s11011-022-01154-7.

Author Response

#1: Changed.
#2: Changed.
#3 & #6: The word "unexpected" has been removed from the manuscript text.
#4: Protocol# is provided in line 108.
#5: Now presented as glutathione disulfide (GSSG, oxidized glutathione)
#7: The references have been carefully reviewed throughout the manuscript and now updated to include publications of more recent years. However, this particular
review paper has not been cited due to its focus on linking diabetes with epilepsy, which is different from our study focus.

Reviewer 3 Report

This manuscript reports for the first time that the combined deficiency of GSH and ascorbate in mice is a direct cause of epileptic seizures leading to premature death — not only in pre-weanling pups, but also in adult rescued mice when dietary AA was removed. It provides potentially an important step to understand the pathophysiology of seizure in humans. The authors should pay attention to some points described below.

Major issues

Measurement of uric acid in HET, MKO, GKO, and DKO must be provided, because this endogenous compound is known to have an equal antioxidant property with AA and to be related to neurodegenerative disorders such as Alzheimer’s disease and Parkinson’s disease. Therefore, some compensatory mechanisms should operate and uric acid concentration may have affected phenotype of each mutant. Since this paper is the original one of DKO, all basic information should be incorporated into it. This also could be the reason for lack of pathological changes in the cerebellum, where uric acid compensated the lack of GSH and AA.

Minor issues

1.      It is well known that AA deficiency causes scurvy. Were there any sings implying for scurvy in AA deficient mice?

2.      Southern blot should be provided in DKO in a supplement datum.

3.      Please explain in the method section the reason for different euthanasia methods being used for biochemical measurement of redox molecule and measurements of hematocrit and plasma glucose and lipid profile.

Author Response

Response to major issue: This is an interesting comment. Unfortunately, we do not have biological samples available to perform the measurement of uric acid. We did incorporate the discussion of potential roles of other antioxidants in our mouse model in this revision (line 435-439).   

Response to minor issues: 

#1: In the original submission, we mentioned that DKO mice did not develop clinical AA deficiency, which refers to scurvy. We have now clarified that statement. 

#2: To confirm the double deletion of Gclm and Gulo genes in DKO mice, we performed PCR genotyping according to protocols that have been published. We now presented these data as Fig. 1B. In addition, we also examined the mRNA levels of the Gcl and Gulo genes, presented as supplemental Fig. 2. Results from these experiments confirmed the disruption of Gclm and Gulo genes and non-detectable messengers of both genes in DKO mice.  

#3: This is a typo. “monoxide” has been changed to “dioxide”.

Reviewer 4 Report

Review of the manuscript entitled: Fatal Epileptic Seizures in Mice Having Compromised Glutathione and Ascorbic Acid Biosynthesis. The manuscript is interesting but many corrections need to be made. In both "summary" and "introduction" the aim of the manuscript is missing. Moreover, Introduction must end with the aim of the paper. Aim of the manuscript should be in line 82 and everything after line 84 should be deleted from introduction. Such information may be include in the summary/conclusions at the end of the manuscript.

The methodology is described correctly.

In the results chapter, the authors mix a description of the results with a discussion. This shows a lack of manuscript writing skills. In the results section, we do not discuss the results and do not analyze them! Please correct the results section. Moreover, we do not add references in the results section! In the results section, we precisely describe the obtained results without interpreting them. All analysis of the results should be in the discussion. If the authors don't know, "discussion" means that we discuss the results and analyze them.

Author Response

#1: We have changed our wording in the last paragraph of Introduction to specifically state the main aim of our study. However, we kept two sentences following the aim, which very briefly introduced the method and key findings. This is in accordance with the Author instructions published on the Journal’s website, which states “......Finally, briefly mention the main aim of the work and highlight the main conclusions.”

#2: We reviewed our paper carefully and have moved some discussion from the Result to the Discussion section. However, in the Result section, we kept some brief discussions of specific data set to facilitate a better interpretation of our data and in some cases to provide rationales for the next experiment. We believe such discussion is important and will benefit the readers. Differently, the discussions presented in the Discussion section are meant for much broader and in-depth discussion of the study as a whole piece and in a biological context that may be beyond the scope of the current study.   

Round 2

Reviewer 1 Report

I don't have any other suggestions. The revised manuscript can be accepted.

Reviewer 2 Report

The authors have conveniently addressed my comments and queries, thus, I recommend to accept the revised version for publication at Antioxidants.

Reviewer 3 Report

n/a

Reviewer 4 Report

The authors took into account my suggestions.